# Experiences of gender-diverse youth during the COVID-19 pandemic in Canada: A longitudinal qualitative study

**Louis Everest[1], Jo Henderson[1,2], Mahalia Dixon[2], Jacqueline Relihan[2], Lisa D. Hawke[1,2]***

**1** University of Toronto, Toronto, Ontario, Canada, **2** Centre for Addiction and Mental Health, Toronto, Ontario, Canada

* Lisa.Hawke@camh.ca

**Data Availability Statement:** Data sharing and confidentiality are governed by the Research Ethics Board of the Centre for Addiction and Mental Health (046-2020). The CAMH Research Ethics

## Abstract

### Background

Recent studies have suggested that the COVID-19 pandemic has increased existing health challenges experienced by transgender (trans) and non-binary people. Additionally, COVID-19 has also negatively impacted youth mental health. However, the impact of the COVID-19 pandemic with respect to the intersection of youth and trans and non-binary populations is currently not well established. The present longitudinal qualitative study aimed to examine the evolving challenges experienced by trans and non-binary youth during the COVID-19 pandemic.

### Methods

Gender-diverse youth, defined as participants who did not identify as cisgender in April 2020, were invited to participate from among the participant pool of a COVID-19 cohort study in Canada. Qualitative interviews were conducted in August 2020, January 2021, and August 2021, during the first year and a half of the COVID-19 pandemic. Qualitative themes were identified based on reflexive thematic analysis and plot-line narrative inquiry.

### Results

Ten participants aged 18 to 28 were included in the present analysis, yielding 29 transcripts. We identified themes of (1) losses of connection to gender-diverse communities, (2) changes in gender identity, affirmation, and self-reflection, (3) a dual burden of trans and non-binary specific health and service access challenges as well as COVID-19 pandemic related health challenges, and (4) virtually rebuilding gender-diverse communities during the COVID-19 pandemic.

### Conclusions

Gender-diverse youth may experience unique challenges during the COVID-19 pandemic. The losses with gender-diverse communities may further isolate queer people from access

Board may be contacted via email at research.ethics@camh.ca. The full dataset may be made available upon request to the corresponding author through a Research Ethics Board application process.

**Funding:** This study was funded by the Canadian Institutes for Health Research (CIHR, #162358), with support from the Margaret and Wallace McCain Centre for Child, Youth and Family Mental Health. The funders had no role in the design or conduct of the research. The funders had no role in the design or conduct of the research.

**Competing interests:** The authors have declared that no competing interests exist.

to healthcare, housing, and employment. Public health policy targeted at gender-diverse youth may consider addressing "upstream" disparities in healthcare and housing in order to support the rebuilding of queer and gender-diverse communities by gender-diverse people.

## Introduction

The coronavirus (COVID-19) pandemic and associated public health measures have led to intensive social changes since COVID-19 was first declared as a pandemic by the World Health Organization on March 11[th], 2020 [1]. Furthermore, the pervasive negative impact of the COVID-19 pandemic on mental health and substance use has generated concerns in the literature with respect to developmental challenges in adolescent and youth populations [2–6]. Recently, health and social disparities in gender-diverse people including transgender (trans) and non-binary people have also been suggested to have increased during the COVID-19 pandemic, based on inequalities derived from high levels of stigma, discrimination, and other adverse structural conditions [7–9]. The intersection of youth and historically marginalized groups may therefore represent a population vulnerable to the social and health challenges associated with the COVID-19 pandemic [5, 10, 11]. However, relatively limited research has examined the qualitative experiences of trans and non-binary youth during the COVID-19 pandemic [6, 8].

Emerging literature examining the experiences of trans and non-binary people during COVID-19 has suggested the pandemic has systemically impacted socioeconomic outcomes, access to healthcare, including gender affirming care (e.g., hormone replacement therapy, gender affirming surgery), as well as social networks, among other domains [5, 6, 8, 9, 12, 13]. Specifically, as examined in a qualitative survey by Kia et al., reductions in access to gender-affirming care, as well as housing and financial insecurity were common experiences for trans and non-binary people in Canada during the COVID-19 pandemic [8]. Further, as examined by Jarrett et al. in a quantitative cross-sectional survey, limited access to gender-affirming care during COVID-19 may have driven increased depressive symptoms, anxiety, and suicidal ideation in trans and non-binary populations [14]. Mittal and Singh (2020) identified an increase in gender-based violence during the early-stages of the COVID-19 pandemic [15]. Further, Hawke et al. identified that trans and gender diverse youth may have experienced larger mental health and substance use burdens during the height of the pandemic, in comparison to cisgender youth [7]. These studies suggest that trans and non-binary people have experienced unique health disparities during COVID-19, and such health disparities may have increased over the pandemic. However, the different endpoints, examined timeframes, and research questions of these studies make identifying trends challenging. Furthermore, in the absence of longitudinal studies examining the experiences of trans and non-binary youth during COVID-19, it is difficult to establish how and why health disparities in trans and non-binary youth may have changed during the pandemic.

Prior literature has suggested that the minority stress model may be applied to trans and non-binary populations to examine the unique stressors they may face (e.g., harassment, healthcare, and employment discrimination) during the pandemic, as well as the adverse health impacts from these stressors [13]. Additionally, as examined by Pease et al., a psychological mediation framework may supplement this conceptualization in gender diverse young adults, where gender dysphoria and emotion dysregulation may be modelled as mediators [16]. In comparison, Diamond and Alley suggest that insufficient social safety may be a primary cause of stigma-related health disparities in gender- and sexually-diverse populations

[17]. In the context of COVID-19, the experience of trans and non-binary youth may be impacted by changes in social support resulting from public health measures such as the closure of in-person schools and financial instability that may require them to return to their parental home. [6, 9, 18]. Therefore, in order to examine how trans and non-binary identities may intersect with experiences during the COVID-19 pandemic, the present study considers an intersectionality theoretical framework.

Recently, calls for longitudinal studies examining trans and non-binary populations during the COVID-19 pandemic that incorporate qualitative methods, have been presented in literature [12, 13, 19, 20]. Furthermore, emerging research suggests that trans and non-binary populations may have experienced a disproportionately increased health burden during COVID-19, compared to cisgender populations [7, 8]. Therefore, the objective of the present longitudinal qualitative study was to examine the experiences of trans and non-binary youth during the first year and a half of the COVID-19 pandemic.

## Methods

### Participants

The present study examined a subset of participants from a larger longitudinal cohort study examining the mental health experiences of youth during the COVID-19 pandemic in Canada, including mental health clinical and non-clinical samples [2]. Detailed inclusion criteria for this study have been previously reported [2, 21]. Briefly, participants were identified based on four existing studies conducted by the Centre for Addiction and Mental Health (CAMH) in Toronto, Ontario, Canada. These studies included three clinical study cohorts recruited from mental health or substance use services at CAMH, and one non-clinical study cohort recruited from schools across Ontario in 2011–2013 [2, 21]. Qualitative interviews were conducted among a subsample of the longitudinal cohort study, of which this study reports on the experience of the gender-diverse participants in the sample. The current sample includes all participants who identified as gender-diverse at the April 2020 survey in the longitudinal cohort study who agreed to participate in qualitative interviews in the August 2020, January 2021, and/or August 2021 data collection waves. For context, in July and August 2020, Ontario, Canada, COVID-19 cases were progressively declining, and the first provincial state of emergency was lifted in conjunction with many public health restrictions [22]. In January 2021, Ontario announced shutdowns, based on rapid increasing case counts [23]. Additionally, the first doses of the Pfizer–BioNTech COVID-19 vaccine were administered to healthcare workers and high-risk populations [24]. In August 2021 Ontario experienced rising case counts; approximately 64% of people in Ontario had received 2 of more COVID-19 vaccine doses [25]. Gender-diverse participants were identified as all participants who did not identify as cisgender at the April 2020 survey. Gender identity was collected based on a combination of list and open text box fields. We identified 10 participants for inclusion; 8 were derived from the clinical sample and 2 from the non-clinical sample. Follow-up in the sample was excellent: 90% of participants provided qualitative interviews at all three timepoints examined, excluding one participant who was missing a qualitative interview at the 2022 August data collection wave. This produced a total of 29 transcripts.

### Procedure

**Participants.** Because of social distancing measures, interviews were conducted over the phone or through video conferencing software (Webex). The interviews had an average length of 78- (August 2020), 73- (January 2021), 87-minutes (August 2021), with a total range of 52-

to 119-minutes. Interviews were digitally recorded and transcribed by a staff member or a professional transcriber, and identifying information was redacted at the transcription stage.

Research Ethics Board approval was obtained through the Centre for Addiction and Mental Health. Electronic written informed consent was obtained for all participants.

## Measures

Quantitative sociodemographic baseline data, including age, education levels, living situations, and region type status, were extracted from the companion quantitative study [2].

Interview topic guide questions were identified by the research team and youth co-researchers based on domain knowledge and prior studies examining the general population of the present cohort. The structure of the interview was designed to be flexible to facilitate open-ended discussions about a range of experiences during the COVID-19 pandemic.

## Youth engagement

The present study was developed based on the Strategy for Patient-Oriented Research [26], using pragmatism as an underlying paradigm [27]. Therefore, this youth-engaged project aimed to examine questions identified by youth co-researchers. Further, qualitative trends and themes were also co-interpreted with youth co-researchers (MD, JR), who co-authored the present manuscript. Engagement was guided by the McCain Model of Youth Engagement [28]. Engagement is reported on using the GRIPP2 checklist [29].

## Reflexivity

As a component of reflexive methodology of the present study, we acknowledge the influence of our experiences and perspectives on the analysis, interpretation, and writing of this report. Our team includes individuals with diverse lived experiences of illness, varying degrees of privilege, power, and vulnerability concerning healthcare access and mental health support. Additionally, our team comprises members with intersecting identity markers such as ethnicity and gender.

## Analysis

The present study utilized the concepts of reflexive thematic analysis [30]. Additionally, we utilized concepts from Bruner's paradigmatic-type plot-line narrative inquiry to examine how themes may change over time [31, 32]. Qualitative data was imported into NVivo 12. A research analyst (LE) read all participants' transcribed interviews and generated initial codes to familiarize themselves with the data. Themes and longitudinal trends were then defined through multiple discussions between LE, youth co-authors (MD, JR), and one of the research leads (LDH). Representative quotes are provided for each theme; participant numbers are sequential and are not linked to study records. Reflexivity and assumptions held by the researchers were discussed throughout the study.

In order to improve the trustworthiness and credibility of our findings, participants had the opportunity to ask questions, revisit previous responses, and amend their responses as facilitated by the interviewer at all follow-up data collection waves. Further, the prolonged engagement with participants and co-interpretation of the results in collaboration with youth with lived experience (MD, JR) support the trustworthiness of our analyses.

## Results

### Study participants

Sociodemographic characteristics of participants included in the present study are summarized in Table 1. At baseline, six participants identified as non-binary, two participants identified as genderqueer, one participant identified as a trans woman, and one participant identified as androgynous. Additionally, 80% of participants were 18- to 22-years, 80% of participants reported they were students at baseline, and 20% of participants identified as racialized.

The results of the present study are organized into four themes, which are organized sequentially to reflect the temporal changes in experiences of trans and non-binary youth included in our study. The examined themes include: (1) losses of connection to gender-diverse communities, (2) changes in gender identity, affirmation, and reflection, (3) dual burden of trans and non-binary specific health and service access challenges as well as pandemic related health challenges (4) rebuilding virtual gender-diverse communities and social networks. The identified themes and supporting quotes are presented in Table 2. Furthermore, critical reflections on engagement based on the GRIPP2 checklist are presented in Table 3.

### Losses of connection to gender-diverse communities

During the August 2020 interviews, participants identified losses of connection to gender-diverse communities based on COVID-19 restrictions, lockdowns, and stay-at-home orders.

**Table 1. Summary of demographic characteristics of included participants.**

| Characteristic* | N = 10[1] |
|---|---|
| **Age** | |
| 18–22 | 8 (80%) |
| 23–28 | 2 (20%) |
| **Gender Identity** | |
| Androgynous | 1 (10%) |
| Genderqueer | 2 (20%) |
| Non-binary | 6 (60%) |
| Trans woman | 1 (10%) |
| **Ethnic Background** | |
| Another Background | 1 (10%) |
| Latin American (e.g., Argentinean, Chilean, Salvadoran) | 1 (10%) |
| Multiple Ethnicities | 1 (10%) |
| White (e.g., English, Canadian, Portuguese) | 7 (70%) |
| **Born in Canada** | |
| No | 1 (10%) |
| Yes | 9 (90%) |
| **Highest Level of Education** | |
| Greater than high school | 7 (70%) |
| High school diploma or less | 3 (30%) |
| **Current Student** | |
| No | 2 (20%) |
| Yes | 8 (80%) |

[1]n (%)

*: Demographic characteristics were collected at baseline (2020).

**Table 2. Thematic analysis summary.**

| Losses of connection to gender-diverse communities | Changes in gender identity, affirmation, and reflection | Dual burden of trans and non-binary specific health and service access challenges as well as pandemic related health challenges | Rebuilding virtual gender-diverse communities and social networks |
|---|---|---|---|
| "I'm looking to move after university, and if the pandemic is still going on, that kind of makes me nervous about prospects about finding kind of queer community in whatever city I end up in." (Participant 1) | "I know all of medicine is backed up right now, but especially for trans people, their surgeries are often labelled as elective or non-essential, which is highly inaccurate. So, yeah, I might just say that people should be aware that these are not elective procedures, and that delaying them has consequences." (Participant 2) | "I know that I'm not the only one who has noticed them put in, like, in more force. I have a lot of housemates who have been called slurs and have noticed them more since the pandemic started than before." (Participant 3) | "I would prefer not being in the pandemic and I'd prefer being able to interact with people in my community and people around me a lot more." (Participant 4) |
| "I'm still living with my parents. Things have gotten bad enough at home because my parents are very transphobic, which I only started realizing within the past two weeks of a lot of things happening that that is the case." (Participant 3) | "So, it's likely the amount of time that I spend around people who don't see me as the gender that I identify as. That's contributing to that, so I don't know if those feelings will stay as things hopefully open back up a little bit more in the future, or if I'll be coming out to them before that happens. " (Participant 6) | "Yeah, I find that being in a virtual setting has made people a lot more comfortable in certain ways. Not in every way, but, you know, the illusion of being behind a screen I think plays into it a little bit, where people feel more emboldened to just kind of say what they want to say, because they know they don't have to deal with any sort of physical like in-person consequences." (Participant 8) | "I feel like that's sort of a big change, is that there's less of that natural sort of safety net that our community can provide by in-person interaction. You have to be a lot more purposeful with it now." (Participant 8) |
| "But it's also not convenient if like, you're not home alone, and you wanna talk about something very intense, so you have to make sure that no one else hears you. So, you kinda have to plan at certain times, otherwise it's sort of easier kind of just being an adult." (Participant 2) | "There are, kind of, intersections at play. Like, I as a white trans woman—I am going to have a much different experience than trans women of color, who are going to be, you know, an insanely vulnerable group in society. Especially, like, during a pandemic when they generally will have, you know, less of a social safety net. And I think, just, increasing, like, public resources in general will, you know, help, like, all marginalized communities and identities and not even just specifically trans women." (Participant 1) | I've been kind of poking around looking for a new family doctor that's trans-informed that I could hopefully speak to—speak about these things to, but again I have not been doing so with really any sense of urgency. I think over the past couple of months, maybe weeks, I've recognized my desire to come out to my family 'cause they're kind of the last stepping stone before I feel like I can just, kind of, be out in the way that I want to generally. So, that's something that I've been thinking a lot about. I haven't taken any steps to make it happen, but that's been a little bit of a point of stress for me is—and I think that it's because I spend so much time at home, and the people that I am out to I don't see in person very often or interact with very much. So it's likely the amount of time that I spend around people who don't see me as the gender that I identify as. (Participant 6) | "I mean, like, in some sense I haven't had the social outlet, like there's been some stuff online, but I think part of it's like going somewhere and having the physical space and closeness to people. (. . .). I mean, not that I'm not myself at home but it's different. I know there's been like Zoom parties and things, but it's not really the same. So, I guess like there's like a little less affirmation and community in that way, because like at home, even if I join one of those things it's not really the same space or privacy I guess; but like later into the pandemic I've also like reconnected or connected with other queer and trans people." (Participant 5) |

Specifically, during the early stages of the pandemic, participants reported concerns that traditional queer spaces (e.g., LGBTQ2S+ community centres, bookstores, support groups, nightclubs, and cafes among others), were no longer accessible. For example, one participant discussed the impact of COVID-19 restrictions on queer spaces, and the impact of this change on social networks:

"I think the pandemic can definitely have some barriers. Just in terms of, you know, finding (. . .) housing, queer spaces—and finding people through those and meeting up and kind of developing relationships that way. Especially because, you know, a lot of the times (. . .)

**Table 3. GRIPP2 reporting checklist for youth engagement in the study.**

| Section and topic | Description |
|---|---|
| 1. Aim | Youth with lived experience were engaged in this study in order to improve the relevance of the interview questions asked, as well as of the research process and interpretations. |
| 2. Methods | The study included multiple youth with lived experience as team members, with intersecting identities of ethnicity, status, and gender. Youth contributed to the design of the parent study and the qualitative sub-study, including the interview guide. The interim results of the study were discussed with two youth with lived experience on two occasions. Feedback from youth was incorporated into all stages of the study. |
| 3. Study results | The study results were informed by the interview guide in which youth feedback was sought during the study design process. In reviewing the findings, youth considered that the results resonated with them and highlighted areas for discussion. Youth emphasized the importance of future long-term studies and studies in racialized populations, with respect to the impact of COVID-19 on gender diverse people. |
| 4. Discussion and conclusions | Members of the research team included youth with lived experience, who contributed to the identification and interpretation of narratives in the present study. We did not experience any negative effects of including youth with lived experience in the present study. |
| 5. Reflections/critical perspective | As the present study was based on a large longitudinal cohort examining quantitative and qualitative outcomes, occurring at a fast pace, we experienced challenges engaging a consistent group of youth over all aspects of the study, throughout the entire study duration. However, access to a broad group of youth through the institutional youth engagement infrastructure facilitated the engagement process and ensured that youth feedback was collected throughout. |

there are kind of spaces, which (. . .) queer people will go to, so those spaces they're taken away. Like, I need to seek those out further avenues."

(Participant 1)

Additionally, during the August 2020 interview, many participants reported that they returned to their parental home, and moved away from urban areas and schools. Some participants identified this as a barrier to accessing gender-diverse and LGBTQ2S+ communities. Specifically, participants discussed the cancelation of in-person events (e.g., 'Pride' events), as well as challenges engaging with their social networks over the phone or video-chat, based on concerns that household members may overhear them disclosing personal information.

"But it's also not convenient if like, you're not home alone, and you wanna talk about something very intense. So, you have to make sure that no one else hears you. So you kind of have to plan at certain times, otherwise it's sort of easier kind of just being an adult."

(Participant 2)

Importantly, some participants reported that privacy concerns in their parental household were based on family members not knowing or accepting their gender identity. For example, one participant reported concerns their laptop and cellphone may be taken away from them by their parents, because of their gender identity.

"I'm hiding a lot of things from them because it's not safe for me to tell them these things—because they're transphobic and they will try to take my money and all of these things just

to hold me from leaving. So, I have to do what's good for me and that means laying low until I can leave."

(Participant 3)

### Changes in gender identity, affirmation, and reflection

During the August 2020 and January 2021 interviews, some participants discussed COVID-19 related public health measures intersecting positively with their gender identity. For example, one participant noted that they "like the androgyny of face masks," (Participant 4) because they were misgendered less frequently while wearing a mask in public. Additionally, several participants reported a positive aspect of the COVID-19 pandemic included increased time to independently reflect upon their own gender identity. Further, several participants reported a change in their gender identity during the duration of the study. Youth also noted that they also observed increased gender reflection in their communities.

"There's been so many people to come out during the pandemic, because you get locked up with yourself for a while and, well—surprise! A lot of people have realised that they're trans. (. . .) I guess, the same thing happened to me, I was not expecting that it would, because I kind of thought I had figured myself out on that front, but yeah."

(Participant 1)

However, participants also discussed negative impacts of COVID-19 related public health measures, based on their gender identity. Specifically, some participants discussed challenges in feeling affirmed in their gender in the absence of public and community affirmation.

"Gender has definitely played a huge part in the time of COVID-19. I am non-binary/a-gender, but I'm also. . ..I consider myself to be a trans-masculine person. (. . .) It's very difficult because it is easier to reassert who you are when there are other people around to perceive you. When you are alone, in your house, it can be more complicated how you assert yourself to yourself because, you see more of who you are than other people."

(Participant 2)

Some participants discussed challenges feeling affirmed in their gender because they were living with only cis-gender family members. These challenges were discussed by participants both with family members who were supportive and non-supportive of their gender identity. Similarly, some participants noted how losses of connection to gender-diverse and queer communities impacted how they felt affirmed with respect to their gender:

"I mean, like, in some sense, I haven't had the social outlet. Like, there's been some stuff online, but I think part of it's like going somewhere and having the physical space and closeness to people (. . .). Not that I'm not myself at home, but it's different. I know there's been like Zoom parties and things, but it's not really the same. So I guess like there's like a little less affirmation and community in that way, because like at home, even if I join one of those things, it's not really the same space or privacy I guess."

(Participant 5)

Some youth also discussed gender-identity reflection intersecting with other aspects of identity. Specifically, youth commented on how social movements and protests, such as race-based discrimination and social disruption [33], impacted how they understood their own gender identity in relation to their race and socioeconomic status.

"The extra time that I have has helped me to get a grasp on different parts of my identity in a way that has been very helpful for how I feel about myself, and how I feel about my place in the world around me."

(Participant 6)

During the present study, gender affirmation and identity were highlighted as an important and meaningful change by many participants. However, as examined in the above section, participants experienced heterogenous impacts of pandemic-related public health measures with respect to positive and negative changes in gender affirmation.

## Dual burden of trans and non-binary–specific health and service access challenges as well as pandemic-related health challenges

Many participants identified mental health challenges that were also reported by cisgender youth during the COVID-19 pandemic. Specifically, substance use, anxiety, and depression were reported by many of the participants. However, on the January 2021 round of interviews, nearly a year into the pandemic, many participants also discussed the impact of the transphobia and discrimination that they experienced on their mental health during the pandemic. As discussed by one participant, increases in transphobia and homophobia heightened feelings of anxiety:

"There's definitely been like more people out and about who like don't agree with things, like who are transphobic or homophobic. And I've noticed them a lot more, because—I don't know why—but like, because of the pandemic they've just come out more and are around more. And it's kind of—it's very unsettling."

(Participant 3)

Additionally, many participants reported increases in misgendering, as well as physical and verbal harassment and assaults during the pandemic. Some youth suggested increases in transphobia may be because in a virtual setting, transphobic "people feel more emboldened to just kind of say what they want to say" (Participant 8). Further, some youth also discussed increases in transphobia and microaggressions from health care providers:

"I would say the discrimination I've experienced has actually mostly been from, you know —what am I trying to say—medical care providers, like from a psychiatrist or something like that. And it's not necessarily violent discrimination. It's more so, okay, you're clearly uncomfortable with trans people, sort of thing, you know. (. . .) But if you feel that there's certain elements of the identity, or you experience that they're not going to understand or that they're going to treat in a similar way to other things that you've sort of witnessed, then you feel less inclined to be honest about your experiences or explore the full range of sort of the care that you could be provided."

(Participant 8)

In addition to concerns regarding gender-based violence, some participants reported challenges with respect to their access to gender-affirming health care. Specifically, youth reported concerns regarding the cancelation of gender-affirming surgeries and the potential unavailability of hormone therapy based on limited physician availability and hormone shortages. Further, many participants discussed feelings of anxiety and frustration at the classification of gender-affirming surgeries as "non-essential," both for themselves and people in their communities. As discussed by one participant, gender-affirming care is an essential part of the lives of trans and non-binary youth with serious impacts on mental health and delaying access to gender-affirming care and surgeries "will have consequences" (Participant 1). One participant discussed their own experiences with the health care system:

> "I was supposed to get breast reduction surgery in June, end of June, cause I wanted to wait until after the school year [. . .] It just got pushed back for like a month. And I emailed them, like, 'Hey, I know that elected surgeries are allowed back in Ontario. Can you let me know when mine is?' And they never responded to me. And I really wanted this. I really, really wanted the surgery, and I don't know what's happening with it now, but I'm just waiting for an indefinite period of time."

(Participant 2)

Participants also reported challenges finding gender-informed health care resources in more rural areas, where they were residing due to the pandemic. These challenges were particularly difficult when seeking mental health supports. As discussed by one participant:

> "But going to a medical hospital in an area that's very rural and doesn't have psychiatric supports in place is a whole other thing. You're not connected to the support that you're looking for, for an overnight the way you would at [psychiatric teaching hospital], which is, it's not viable."

(Participant 3)

### Rebuilding virtual gender-diverse communities and social networks

During the August 2021 round of interviews, about a year and a half into the pandemic, youth reported the process of rebuilding gender-diverse and LGBTQ+ communities. Youth identified rebuilding gender-diverse communities through a plurality of modalities. Specifically, in-person, as well as video-, voice-, and text-based chats social networks were discussed. Many youth discussed the importance of queer spaces with respect to developing social connections, finding safe job opportunities, as well as gender-educated health care professionals. Furthermore, as discussed by one participant, the serendipitous "drop-in" nature of social networks prior to the pandemic were reflected in the rebuilt online queer spaces:

> "I just think it's important to be able to connect with people who—I can do online groups and drop in groups with people who identify as non binary and trans. And go to drop in groups and events. There's the drop-in that I used to go to and stuff like that."

(Participant 3)

Furthermore, some participants found queer communities in online role-playing groups. Youth reported that these groups may facilitate gender-exploration and provided a sense of

autonomy with respect to one's gender and environment. Additionally, one participant discussed accessing online communities with respect to their spirituality:

"I have been looking for a community to sort of connect to more on a permanent basis, but it's been difficult with the pandemic and with other circumstances as well. So I've occasionally connected to online spaces that have been really helpful in fostering sort of my sense of connection and community in that way, to my spirituality"

(Participant 8)

Participants also discussed how the shift to virtual communication facilitated the reconnection of some queer communities, now geographically separated. However, some youth noted that such rebuilt social networks did not eliminate feelings of isolation:

"Later into the pandemic, I've also like reconnected or connected with other queer and trans people and like, groups of people that like I had talked to briefly, but like we'd add each other on Facebook and not really talked in depth. And now I have like a really good friend in the States—and right now not so much, cause I was away—but we had been calling each other twice a week and checking in and stuff and that was really nice. So I guess like I've found ways to get around it, but that has felt kind of isolating and challenging I guess, to not have those physical gatherings and spaces."

(Participant 5)

Importantly, we also identified that not all youth benefited equally from these online communities, and disparities in access to gender-diverse communities still exist. Additionally, substantial heterogeneity existed between youth with respect to their preference of community mediums. Specifically, some youth who identified as being early in their gender-transition reported not feeling comfortable seeing their face in a video-chat. In comparison, other youth discussed not being able to adequately communicate feelings and make connections through text-based chat. Additionally, many youth discussed insufficient privacy as a barrier to access online gender-diverse communities.

## Longitudinal changes

As examined in Fig 1, we suggest that the identified themes are connected, and relate to specific periods and social events during the pandemic. Specifically, we suggest that the losses of traditional queer spaces impacted gender identity because youth did not have access to community social supports. Participants discussed these challenges as most impacting their lived experience during the first provincial state of emergency in spring/ summer of 2020.

We suggest that the cohort of youth who experienced changes in gender identity during the pandemic experienced unique traumas in the absence of queer support communities. Therefore, the need for youth in this cohort to seek out help from other gender diverse people, and the desire to support other gender diverse people experiencing trauma, naturally drove the development of new gender diverse and queer communities. Furthermore, queer people recognizing the gaps in access to housing, employment, and healthcare were a driving force of the observed rebuilding of virtual gender-diverse communities. These changes were identified later in the pandemic, during the January 2021 and August 2021 interviews.

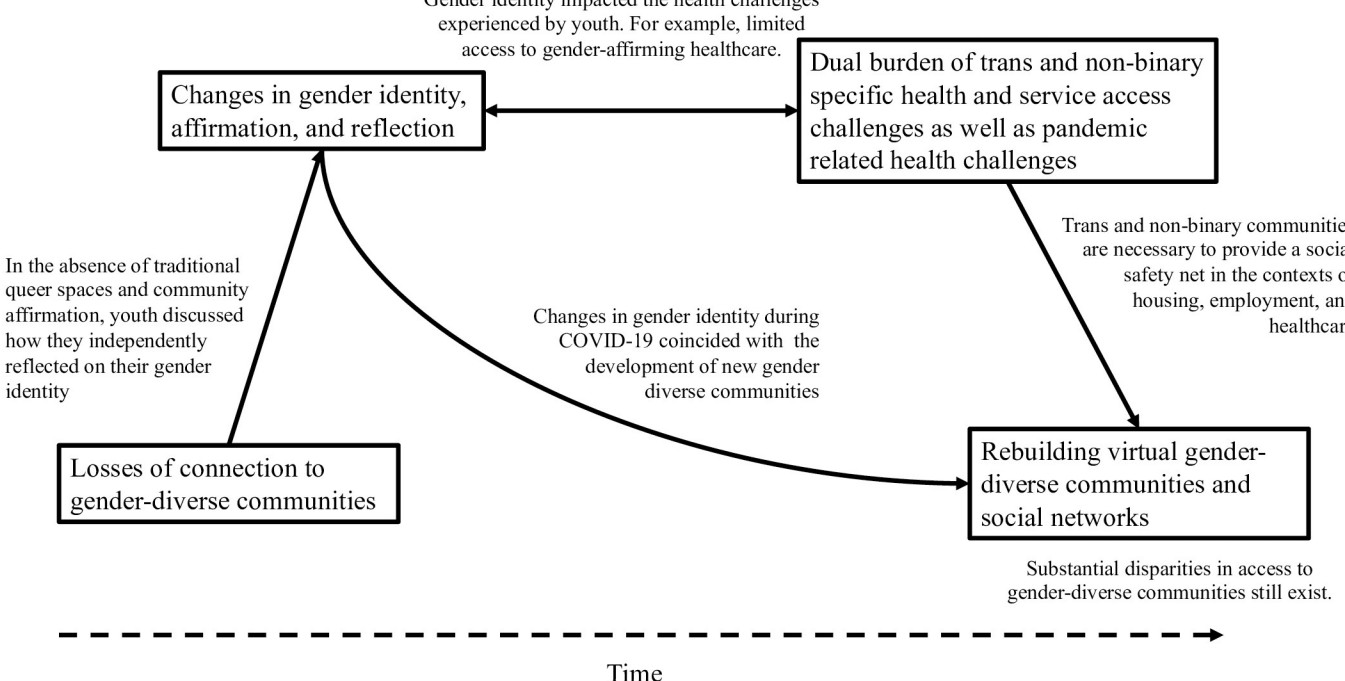

**Fig 1. Summary of examined themes.** Legend: Boxes represents examined themes; solid lines represent connections between themes; dashed lines represent changes in time from the start of the COVID-19 pandemic.

## Discussion

In the present qualitative longitudinal cohort study examining the impact of COVID-19 on gender-diverse youth populations in Canada, we identified (1) loss of connection to gender-diverse communities, (2) changes in gender identity, affirmation, and reflection, (3) a dual burden of trans and non-binary specific health and service access challenges as well as pandemic related health challenges, and (4) rebuilding virtual gender-diverse communities during the COVID-19 pandemic. Importantly, gender-diverse communities provide a "safety net" for trans and non-binary youth that may facilitate the informal exchange of healthcare, housing, and employment information between trans and non-binary youth; such safety nets are essential due to the continuing experiences of institutional transphobia [34]. The present study contributes to the growing body of research examining the unique and specific challenges that trans and non-binary youth may experience, and how these challenges may have changed over the COVID-19 pandemic.

The challenges experienced by gender-diverse young people during COVID-19 in the present study were generally consistent with those identified in prior literature. Specifically, similar to the identified loss of connection to gender-diverse communities and changes in gender identity, affirmation, and reflection, a qualitative study of queer and trans youth in the U.S. identified feelings of isolation, reduced social support, and the importance of queer communities and resources during the first 4-months of the COVID-19 pandemic [6]. Additionally, a U.S. qualitative study of trans and non-binary peer supporters identified the importance of gender-diverse communities with respect to healing shared trauma during the COVID-19 pandemic [34]. Furthermore, similar to the identified loss of connection to gender-diverse communities and changes in gender identity, affirmation, and reflection, Kia et al. identified

changes in gender-diverse social networks based on physical distancing and the shift to virtual communication, as well as changes in gender affirmation and barriers to healthcare during the COVID-19 pandemic in Canada [8]. The present analysis contextualizes this prior literature based on the novel, to our knowledge, identified trend of increased feelings of isolation and loneliness during the first year of the pandemic, followed by reductions in feelings of isolation in conjunction with the rebuilding virtual gender-diverse communities. The present analysis also identified potentially gender-diverse youth–specific challenges that may be underrepresented in present literature, including the impact of moving back to a parental home with transphobic or unsupportive family members following lockdown orders.

The identified dual burden of trans and non-binary specific health and service access challenges, together with pandemic-related health challenges, suggests that a psychological mediation framework may be applicable to examine the reasons trans and non-binary youth experienced the challenges discussed in the present study [16, 35]. Specifically, based on a psychological mediation framework, losses of connection to gender-diverse communities may be mediating the impact of the dual burden of experiences with respect to access to healthcare [35]. For example, gender-diverse communities may facilitate access to gender-affirming care by creating networks and sharing personal knowledge of gender-informed health care practitioners [20]. Therefore, COVID-19 may have directly resulted in healthcare challenges for non-binary and transgender youth (e.g., cancellation of gender affirming surgeries). It may have also indirectly resulted in healthcare challenges based on the access of trans and non-binary youth to gender affirming care being mediated by community supports that were disrupted during COVID-19. Furthermore, the observed heterogeneity with respect to COVID-19 pandemic-related healthcare challenges may be partially based on the differences in gender-diverse community support and access [6, 20]. Specifically, gender-diverse community support and access may be reduced based on intersections between trans and non-binary identities and socio-demographic identities such as ethnicity, geography, socio-economic status, familial and social support among others. Therefore, gender-diverse people with limited access to social and community support may have been most negatively impacted by the COVID-19 pandemic [8]. Additionally, this observation may highlight the need for health policy solutions developed with an intersectional perspective, in contrast to one-size-fits-all solutions.

Trans and non-binary youth represent populations with numerous and substantial existing barriers to healthcare, even prior to the pandemic [7, 8]. Therefore, the challenges examined in the present study may be interpreted as compounding additions to existing disparities in access to healthcare. Furthermore, gender-diverse youth may experience a dual mental health burden, where they may experience unique gender-based pandemic-related and -exacerbated challenges, as well as pandemic-related mental health challenges also observed in cisgender youth, such as increases depression, anxiety, and substance use [7, 34]. Furthermore, trans and non-binary youth may experience a plurality of systemic and institutional barriers in building social networks and connecting with peers [8, 36]. From this perspective, the rebuilding of virtual gender diverse communities may be interpreted as gender-diverse community resilience, as well as the individual resilience of gender-diverse peoples. However, the heterogeneity of experiences in online communities identified in the present narrative highlight the unmet needs of some trans and non-binary youth with respect to rebuilding communities following COVID-19 public health related restrictions.

The present study has several potential applications with respect to the field of public health policy in Canada. Specifically, our study suggests that existing health disparities in trans and non-binary people may have been exacerbated during the pandemic partially based on reductions in capacity for gender-diverse peer support within communities, and the loss of informal community safety nets [34]. However, as examined by Ghabrial et al., community groups

funded by the Canadian government aimed at increasing COVID-19 vaccination rates, may have inadequately addressed pre-existing health inequities and intersecting systems of oppression for gender-diverse peoples during the pandemic [37]. Furthermore, the results support the observations of Kia et al., that COVID-19-related policies in Canada, such as Canada's poverty reduction strategy, do not adequately address the challenges experienced by trans and non-binary people during the pandemic [8]. Therefore, the present study contributes to the existing body of literature suggesting the need for public health policies addressing the needs of trans and non-binary populations, with respect to the domains of healthcare and housing, in order to facilitate the development of peer-led communities. Additionally, the results highlight the importance of the engagement and inclusion of trans youth voices and perspectives when developing public health policies [28, 38]. The current observations may also be applicable to future pandemics, as well as periods of rapid social, economic, and health changes with potential impacts on gender-diverse populations. Importantly, future research should examine the impact of increasing hate and violence towards trans and non-binary peoples during the pandemic, as discussed by some participants in the present study. In the present study, none of the participants identified as Two-Spirit, Indigenous, Métis, or Inuit. However, as discussed by Sylliboy et al., this population experienced unique challenges during COVID-19. Therefore, the experience of Two-Spirit individuals during COVID-19 represents an important area for future research [39].

This study had a number of strengths. Notably, it was a youth-engaged study that benefited from lived experience voices throughout all study stages. It also consisted of in-depth interviews that were conducted longitudinally during a critical time in global public health. An additional strength includes the longitudinal design of the present study, as it facilitated discussion with respect to evolving gender-identities over the pandemic, which may not be captured in a cross-sectional study. However, certain limitations should be kept in mind. The sample size is small, potentially limiting the breadth of the findings. In addition, only 20% of our participants identified as racialized, therefore some perspectives of racialized youth may have been missed. Specific challenges may be experienced by youth with specific gender identities in intersection with various sociodemographic characteristics and social locations, which merits future research.

## Conclusions

Trans and non-binary youth experienced evolving challenges during the COVID-19 pandemic, including losses with gender diverse communities, changes in gender identity and affirmation, as well as the dual burden of trans and non-binary specific health and service access challenges and pandemic-related health challenges. Importantly, the present study identified narratives of resilience and the rebuilding of gender-diverse communities through virtual social networks. In order to support the rebuilding of important gender-diverse communities, public health policy should address the needs expressed by gender-diverse youth, particularly considering community building, healthcare access, and housing disparities.

## Author Contributions

**Conceptualization:** Jo Henderson, Mahalia Dixon, Jacqueline Relihan, Lisa D. Hawke.

**Formal analysis:** Louis Everest, Mahalia Dixon, Jacqueline Relihan, Lisa D. Hawke.

**Funding acquisition:** Jo Henderson, Lisa D. Hawke.

**Investigation:** Jo Henderson.

**Methodology:** Louis Everest, Jo Henderson, Mahalia Dixon, Jacqueline Relihan, Lisa D. Hawke.

**Supervision:** Lisa D. Hawke.

**Validation:** Jo Henderson.

**Writing – original draft:** Louis Everest.

**Writing – review & editing:** Jo Henderson, Mahalia Dixon, Jacqueline Relihan, Lisa D. Hawke.

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
