## [Decision Letter · Decision Letter 0]

13 Sep 2023

PONE-D-23-19615Experiences of Gender-Diverse Youth During the COVID-19 Pandemic in Canada: A Longitudinal Qualitative StudyPLOS ONE

Dear Dr. Hawke,

Thank you for submitting your manuscript to PLOS ONE. After careful consideration, we feel that it has merit but does not fully meet PLOS ONE’s publication criteria as it currently stands. Therefore, we invite you to submit a revised version of the manuscript that addresses the points raised during the review process.

The Reviewer has made some minor suggestions that could improve your article's impact. I especially encourage you to follow their recommendation of discussing the Canadian context (supported by Canadian literature) for gender-diverse youth.

We look forward to receiving your revised manuscript.

Kind regards,

Ietza Bojorquez, Ph.D.

Academic Editor

PLOS ONE

Journal Requirements:

"This study was funded by the Canadian Institutes for Health Research (CIHR, #162358), with support from the Margaret and Wallace McCain Centre for Child, Youth and Family Mental Health. The funders had no role in the design or conduct of the research."

Reviewers' comments:

Reviewer's Responses to Questions

**Comments to the Author**

1. Is the manuscript technically sound, and do the data support the conclusions?

Reviewer #1: Yes

2. Has the statistical analysis been performed appropriately and rigorously? 

Reviewer #1: N/A

3. Have the authors made all data underlying the findings in their manuscript fully available?

Reviewer #1: Yes

4. Is the manuscript presented in an intelligible fashion and written in standard English?

Reviewer #1: Yes

5. Review Comments to the Author

Reviewer #1: Thank you for the opportunity to review your submission. The topic area is of importance, and there is potential for a meaningful contribution to the literature Below are some considerations for the author and editor.

Introduction:

- Well written, clear, and thorough review of the literature

Methods:

- It would be helpful to know more about what province(s) and/ or territory/ies this research took place in. Additionally, it may be beneficial to provide some context into public health restrictions and/or the COVID-19 context during data collection. This will help contextualize experiences and changes.

- Were any techniques used to enhance trustworthiness and credibility of the data analysis (i.e., member checking).

Results:

- Overall, the results section was well written with adequate information figures, and tables. A suggestion would be to add some demographic details to who the quotes are from to add more context. For example instead of just Participant 1 it may say Participant 1, Gender, Age.

- The inclusion of figure 1 was helpful in conceptualizing the results. I would like to see more discussion on the longitudinal context.

Discussion:

- Overall well written. However, given the Canadian context of this study I would recommend connecting the study findings to some of the Canadian literature available on 2SLGBTQ+ populations and COVID-19.

6. PLOS authors have the option to publish the peer review history of their article (what does this mean?). If published, this will include your full peer review and any attached files.

Reviewer #1: No

---

## [Author Response · Author response to Decision Letter 0]

24 Oct 2023

Response to Reviewers 

Dear PLOS ONE Editorial Office,

Thank you for the opportunity to resubmit our manuscript. We are grateful for the comments by the Reviewers which help strengthen our manuscript. Below you will find the point-by-point responses to each of the Reviewers comments. The page and paragraph references refer to the track changes version of the manuscript attached.

Manuscript Title: Experiences of Gender-Diverse Youth During the COVID-19 Pandemic in Canada: A Longitudinal Qualitative Study

Reviewer 1 Comments 

Thank you for the opportunity to review your submission. The topic area is of importance, and there is potential for a meaningful contribution to the literature Below are some considerations for the author and editor.

1. [Methods] It would be helpful to know more about what province(s) and/ or territory/ies this research took place in. Additionally, it may be beneficial to provide some context into public health restrictions and/or the COVID-19 context during data collection. This will help contextualize experiences and changes.

We agree and thank the Reviewer for this insightful comment. In our revised manuscript we have included the following paragraphs to the Methods section: 

“Briefly, participants were identified based on four existing studies conducted by the Centre for Addiction and Mental Health (CAMH) in Toronto, Ontario, Canada. These studies included three clinical study cohorts recruited from mental health or substance use services at CAMH, and one non-clinical study cohort recruited from schools across Ontario in 2011 – 2013.” 

(page 6, paragraph 2)

“For context, In July and August 2020, Ontario, Canada COVID-19 cases were progressively declining, and the first provincial state of emergency was lifted in conjunction with many public health restrictions. In January 2021, Ontario announced a more shutdowns, based on rapid increasing case counts. Additionally, the first doses of the Pfizer–BioNTech COVID-19 vaccine were administered to healthcare workers and high-risk populations. Further, in August 2021 Ontario experienced rising case counts; approximately 64% of people in Ontario had received 2 of more COVID-19 vaccine doses.” 

(page 6, paragraph 2)

2. [Methods] Were any techniques used to enhance trustworthiness and credibility of the data analysis (i.e., member checking).

We thank the Reviewer for this comment. We also agree this is an important component of the methodology of the present study. Therefore, we have included the following sentence to the methods section of the revised manuscript: 

“In order to improve the trustworthiness and credibility of our findings, participants had the opportunity to ask questions, revisit previous responses, and amend their responses as facilitated by the interviewer at all follow-up data collection waves. Further, the prolonged engagement with participants and co-interpretation of the results in collaboration with youth with lived experience (MD, JR) may support the trustworthiness of our analyses.” 

(page 9, paragraph 2)

3. [Results] Overall, the results section was well written with adequate information figures, and tables. A suggestion would be to add some demographic details to who the quotes are from to add more context. For example instead of just Participant 1 it may say Participant 1, Gender, Age.

We thank the Reviewer for this comment. We also agree that including gender identity and age would provide useful additional context for readers. However, given the small sample size of the present study, we have ethical concerns that this may increase the risk of identification for the participants. Importantly, because our study examined only gender diverse Canadian youth in Ontario (in contrast to related qualitative literature examining LGBTQIA+ youth and adults), we hope that the study design may provide adequate demographic context for readers, while still preserving the anonymity of the participants. Therefore, while we thank the Reviewer for this comment, we suggest that the quotes be maintained without specific demographic information. 

4. [Results] The inclusion of figure 1 was helpful in conceptualizing the results. I would like to see more discussion on the longitudinal context.

We thank the Reviewer for this comment. We also agree that is this an important component of the present study. Therefore, we have added the following subsection to the results, under the heading “Longitudinal Changes”: 

“As illustrated in Figure 1, we suggest the identified themes are connected, and relate to specific periods and social events during the pandemic. Specifically, we suggest that the losses of traditional queer spaces impacted gender identity because youth did not have access to community social supports. Participants discussed these challenges as most impacting their lived experience during the first provincial state of emergency in spring/summer of 2020. 

We suggest that the cohort of youth who experienced changes in gender identity during the pandemic experienced unique traumas in the absence of queer support communities. Therefore, the need for youth in this cohort to seek out help from other gender diverse people, and the desire to support other gender diverse people experiencing trauma, naturally drove the development of new gender diverse and queer communities. Further, queer people recognizing the gaps in access to housing, employment, and healthcare were a driving force of the observed rebuilding of virtual gender-diverse communities. These changes were identified later in the pandemic, during the January 2021 and August 2021 interviews.”

(page 18, paragraph 1)

5. [Discussion] Overall well written. However, given the Canadian context of this study I would recommend connecting the study findings to some of the Canadian literature available on 2SLGBTQ+ populations and COVID-19. 

We agree and thank the Reviewer for this insightful comment. In our revised manuscript, we have revised the discussion section to include the following paragraph examining potential connections with respect to Canadian 2SLGBTQ+ and COVID-19 literature: 

 “The present study has several potential applications with respect to the field of public health policy in Canada. Specifically, our study suggests that existing health disparities in trans and non-binary people may have been exacerbated during the pandemic partially based on reductions in capacity for gender-diverse peer support within communities, and the loss of informal community safety nets (32). However, as examined by Ghabrial et al., community groups funded by the Canadian government aimed at increasing COVID-19 vaccination rates, may have inadequately addressed pre-existing health inequities and intersecting systems of oppression for gender-diverse peoples during the pandemic (37). Further, the results support the observations of Kia et al., that COVID-19-related policies in Canada, such as Canada’s poverty reduction strategy, do not adequately address the challenges experienced by trans and non-binary people during the pandemic (8).”

(page 21 paragraph 2)

“In the present study, none of the participants identified as Two-Spirit, Indigenous, Métis, or Inuit. However, as discussed by Sylliboy et al., this population experienced unique challenges during COVID-19. Therefore, the experience of Two-Spirit individuals during COVID-19 represents an important area for future research.”

(page 22, paragraph 1).

---

## [Editor Report · Decision Letter 1]

31 Oct 2023

Experiences of Gender-Diverse Youth During the COVID-19 Pandemic in Canada: A Longitudinal Qualitative Study

PONE-D-23-19615R1

Dear Dr. Hawke,

We’re pleased to inform you that your manuscript has been judged scientifically suitable for publication and will be formally accepted for publication once it meets all outstanding technical requirements.

Kind regards,

Ietza Bojorquez, Ph.D.

Academic Editor

PLOS ONE
---

## [Editor Report · Acceptance letter]

7 Nov 2023

PONE-D-23-19615R1 

Experiences of Gender-Diverse Youth During the COVID-19 Pandemic in Canada: A Longitudinal Qualitative Study 

Dear Dr. Hawke:

I'm pleased to inform you that your manuscript has been deemed suitable for publication in PLOS ONE. Congratulations! Your manuscript is now with our production department. 

Kind regards, 

on behalf of

Dr. Ietza Bojorquez 

Academic Editor

PLOS ONE